# ADAMTSL2 is a potential prognostic biomarker and immunotherapeutic target for colorectal cancer: Bioinformatic analysis and experimental verification

**Zhe Huang** ⓘ *, **Xu Hu, Yiqiu Wei, Yousheng Lai, Jiaming Qi, Jinglin Pang, Kang Huang, Huagui Li, Pengzhu Cai** *

Department of Anorectal Surgery, Affiliated Hospital of Guangdong Medical University, Zhanjiang, Guangdong, China

* 278843749@qq.com (PC); 396018618@qq.com (ZH)

**Data Availability Statement:** The datasets for this study can be found in TCGA-CRC (https://portal.gdc.cancer.gov/) and GSE71187 (https://www.

## Abstract

The ADAMTS Like 2 (ADAMTSL2) mutation has been identified to be associated with different human genetic diseases. The role of ADAMTSL2 is unclear in colorectal cancer (CRC). The study investigated the expression of ADAMTSL2 in both pan cancer and CRC, using data from The Cancer Genome Atlas (TCGA) database to assess its diagnostic value. The study examined the correlation between ADAMTSL2 expression levels and clinical characteristics, as well as prognosis in CRC. The study explored potential regulatory networks involving ADAMTSL2, including its association with immune infiltration, immune checkpoint genes, tumor mutational burden (TMB) / microsatellite instability (MSI), tumor stemness index (mRNAsi), and drug sensitivity in CRC. ADAMTSL2 expression was validated using GSE71187 and quantitative real-time PCR (qRT-PCR). ADAMTSL2 was aberrantly expressed in pan cancer and CRC. An increased level of ADAMTSL2 expression in patients with CRC was significantly associated with the pathologic N stage ($p < 0.001$), pathologic stage ($p < 0.001$), age ($p < 0.001$), histological type ($p < 0.001$), and neoplasm type ($p = 0.001$). The high expression of ADAMTSL2 in patients with CRC was found to be significantly associated with a poorer overall survival (OS) (HR: 1.67; 95% CI: 1.18–2.38; $p = 0.004$), progression-free survival (PFS) (HR: 1.55; 95% CI: 1.14–2.11; $p = 0.005$) and disease-specific survival (DSS) (HR: 1.83; 95% CI: 1.16–2.89; $p = 0.010$). The expression of ADAMTSL2 in patients with CRC ($p = 0.009$) was identified as an independent prognostic determinant. ADAMTSL2 was associated with extracellular matrix receptor (ECM-receptor) interaction, transforming growth factor β (TGF-β) signaling pathway, and more. ADAMTSL2 expression was correlated with immune infiltration, immune checkpoint genes, TMB / MSI and mRNAsi in CRC. ADAMTSL2 expression was significantly and negatively correlated with 1-BET-762, Trametinib, and WZ3105 in CRC. ADAMTSL2 was significantly upregulated in CRC cell lines. The high expression of ADAMTSL2 is significantly correlated with lower OS and immune infiltration of CRC. ADAMTSL2 may be a potential prognostic biomarker and immunotherapeutic target for CRC patients.

ncbi.nlm.nih.gov/geo/query/acc.cgi?acc=
GSE71187). The details of the code and other
original data were deposited on GitHub (https://
github.com/ZheHuang396018618/ADAMTSL2).

**Funding:** This study was supported by the
Competitive Allocation Project of Special Funds for
Science and Technology Development by
Zhanjiang City in 2021 (No. 2021A05080 to Z.H.),
the Clinical Research Project funded by the
Affiliated Hospital of Guangdong Medical University
in 2019 (No. LCYJ2019B007 to Z.H.). The funders
had no role in study design, data collection and
analysis, decision to publish, or preparation of the
manuscript.

**Competing interests:** The authors have declared
that no competing interests exist.

## Introduction

Colorectal cancer (CRC), which affects the colon and rectum, is the third most prevalent cause of cancer-related mortality globally, accounting for more than 1.85 million incidences and 850,000 fatalities annually [1]. These findings align with data from the National Cancer Center, which reports a 5-year survival rate of approximately 50% for CRC in China [2]. CRC is a heterogeneous disorder with different pathogenic mechanisms that involve somatic mutations, gene fusion, genetic instability, and epigenetic alterations [3]. Despite the tremendous efforts made in the treatment of CRC in terms of early diagnosis and multi-disciplinary treatment, including improved screening methods, surgical procedures, chemotherapy, radiation therapy, targeted biologic therapy, and immunotherapy, a significant proportion of patients with CRC, especially those with advanced CRC, still have a poor prognosis [4]. The 5-year survival rate for patients with early-stage CRC is approximately 90%, and once a patient is diagnosed with advanced CRC, the survival rate drops to 13.1% [5]. With the development of genome sequencing technology, increasing biomarkers that predict tumor development and prognosis are being discovered. Therefore, it is necessary to explore potential prognostic biomarkers that facilitate the early diagnosis and treatment of CRC.

ADAMTS Like 2 (ADAMTSL2) is a protein coding gene. This gene is responsible for encoding a constituent of ADAMTS (a disintegrin and metalloproteinase with thrombospondin motifs) and the ADAMTS-like protein family [6, 7]. Mutations in ADAMTSL2 have been found in different human genetic disorders [8–10]. Univariate analysis showed that ADAMTSL2 expression was correlated with the prognosis of hepatocellular carcinoma (HCC) [11]. However, the clinical implications and regulatory network of ADAMTSL2 in patients with CRC remain uncertain.

The objective of this study was to investigate the different expression patterns of ADAMTSL2 in pan cancer and CRC, and to assess the diagnostic importance of ADAMTSL2 in CRC using data from The Cancer Genome Atlas (TCGA) database. Furthermore, we examine the association between ADAMTSL2 expression levels and clinical characteristics, as well as prognosis in CRC. To elucidate the potential regulatory network of ADAMTSL2, genomic enrichment analysis (GSEA) was employed. Furthermore, we investigate the correlation between ADAMTSL2 and immune infiltration, immune checkpoint genes, tumor mutation burden (TMB), microsatellite instability (MSI), tumor stemness index (mRNAsi), and drug sensitivity in CRC. Additionally, we utilized the Gene Expression Omnibus (GEO) database (GSE71187) to validate the abnormal expression of ADAMTSL2. This study has identified a potentially valuable prognostic biomarker and immunotherapeutic target for patients with CRC.

## Materials and methods

The data utilized in this study were sourced from the TCGA and GEO databases. Given that these databases provide free access to data for research and publication purposes, our study did not require the ethical approval or consent of the patient. It is important to note that this study used only cell lines and did not involve the use of human or animal tissues.

### Sample collection

Pan-cancer patients were collected from TCGA database [12]. A total of 647 CRC tissues and 51 adjacent normal tissues were obtained from patients with CRC (colon & rectal cancer,

COAD&READ) [13, 14]. Furthermore, 157 CRC tissues and 32 normal tissues were collected from the GSE71187 dataset.

## Data processing

RNAseq data and clinical data in level 3 HTSeq-FPKM format were collected from the TCGA project [15]. The molecule was ADAMTSL2 [ENSG00000197859.11].

## Differential expression of ADAMTSL2

The R software, ggplot2, and statistical methods (stats [4.2.1] and car) were used for the analysis of the unpaired, paired, and pan-cancer sample.

ROC analysis was performed using the R software and the pROC package, as well as the ggplot2 package [16]. Clinical variables compared tumor and normal samples.

## Correlation of ADAMTSL2 with clinical characteristics

We used the R software and ggplot2 to explore the correlation between ADAMTSL2 expression and clinical characteristics. Clinical variables included N stage, pathologic stage, age, histological type, and neoplasm type.

The R software and the dichotomous logistic model were used for logistics analysis [17]. The dependent variable was ADAMTSL2. The types of independent variables were low and high dichotomous.

## Correlation between ADAMTSL2 and prognosis

Kaplan-Meier analysis was performed using R software, specifically using the survminer package [3.3.1] and the survival package [3.3.6] [18]. We divided the expression levels of ADAMTSL2 into high and low groups. The subgroups were classified into two categories, namely 0–50 and 50–100. The types of prognosis considered were overall survival (OS) and disease-specific survival (DSS). Further prognostic data were obtained from reference [19].

Cox regression analysis was performed using the R software, using the survivor package and the Cox regression module. The prognosis of interest was OS. The clinical characteristics of the variables and ADAMTSL2 were included in the analysis, together with additional prognostic data obtained from a reference source [19].

The R software and the ggplot2 package were used for the forest plot.

The rms package & survival package and Cox were used for the nomogram plot. The prognosis considered in this study was OS. Variables taken into account were T stage, N stage, pathological stage, age, and ADAMTSL2. We collect prognostic data from the reference [19].

## ADAMTSL2-associated pathways

The R software and deseq2 were used for differential analysis of a single gene [20]. The high and low expression groups were defined as 0–50% and 50–100%, respectively.

We used R software in conjunction with the ggplot2 and clusterProfiler packages to perform GSEA [21, 22]. The species selected for the analysis was Homo sapiens, and the reference gene collection used was c2.cp.v7.2.symbols.gmt [Curated]. The gene set database used was MSigDB Collections, which can be accessed through the provided hyperlink and includes comprehensive descriptions of each individual gene set. The significance conditions were generally as follows: P.adj < 0.05 & FDR < 0.25.

## Correlation of ADAMTSL2 with immune infiltration and immune checkpoint-related genes

The immunocell algorithm was applied using ssGSEA, which is an algorithm integrated within the GSVA package [23, 24]. We obtained 24 markers of immune cells from the reference [25]. The statistical method is Spearman.

After grouping the main variables, we select appropriate statistical methods (stats package and car package) based on the characteristics of the data format for statistical analysis (if the statistical requirements are not met, statistical analysis will not be conducted) and use the ggplot2 package to visualize the data. We calculate the stromal and immune scores of the corresponding cloud data using R package estimate [1.0.13] [26]. The statistical method is the Wilcoxon rank sum test.

The RNAseq data (level 3) and the corresponding clinical information of CRC were obtained from TCGA. In this study, the expression of eight immune checkpoint genes, namely, sialic acid binding Ig like lectin 15 (SIGLEC15), Indoleamine 2,3-dioxygenase 1 (IDO1), CD274, Hepatitis A virus cellular receptor 2 (HAVCR2), programmed cell death 1 (PDCD1), cytotoxic T-lymphocyte associated protein 4 (CTLA4), lymphocyte activating 3 (LAG3), and programmed cell death 1 ligand 2 (PDCD1LG2), was analyzed. We analyze the expression of these eight genes. We extracted the expression values of these 8 genes and observed the expression of genes related to immune checkpoint. We used statistical analysis using R software v4.0.3. If not specified otherwise, the rank sum test detects the difference between the two groups of data, and a P-value<0.05 is considered statistically significant.

## The relationship between ADAMTSL2 and TMB/MSI

TMB, an abbreviation for tumor mutational burden, serves as a quantifiable metric for the number of mutations observed in cancer [27, 28]. MSI is a phenotype that occurs due to malfunction of DNA repair mechanisms and is present in approximately 15% of CRC [29]. RNAseq data (level 3) and the corresponding clinical information for CRC were obtained from TCGA. We used Spearman's correlation analysis to describe the correlation between quantitative variables without a normal distribution. A p-value less than 0.05 is considered statistically significant.

## The relationship between ADAMTSL2 and mRNAsi

The OCLR algorithm is used to calculate mRNAsi [30]. Based on mRNA expression characteristics that include a gene expression profile containing 11774 genes, we used the same spearman correlation (RNA expression data) and then mapped the stemness index to the range of [0,1] using a linear transformation by subtracting the minimum and dividing by the maximum [31]. All the above analysis methods and the R software packages were executed using the version v4.0.3 of the R software (R Foundation for Statistical Computing, 2020).

## Expression of ADAMTSL2 in CRC single cells

The Tumor Immune Single Cell Hub 2 (TISCH2) (http://tisch.comp-genomics.org/) database is a scRNA-seq database that focuses primarily on the tumor microenvironment (TME). TISCH2 offers a comprehensive cell-type annotation at the single-cell level, facilitating the investigation of TME in various types of cancer.

## Genomic variants of ADAMTSL2 in CRC patients

The RNAseq data (level3), the mutation maf data and the corresponding clinical information for CRC were obtained from the TCGA dataset (https://portal.gdc.com). Somatic mutations in

CRC patients were downloaded and visualized using the maftools package in R software. Horizontal histograms showed a high frequency of mutations in patients with CRC.

### Drug sensitivity of ADAMTSL2 in pan cancer

ADAMTSL2 drug sensitivity was analyzed in pan cancer using the RNAactDrug database available at http://bio-bigdata.hrbmu.edu.cn/RNAactDrug/index.jsp.

### Validation of ADAMTSL2 gene expression in GEO database

In order to improve the reliability of the TCGA database, data pertaining to CRC samples were obtained from the GEO database. GSE71187 contained 157 CRC tissues and 32 normal tissues. We used GSE71187 for the analysis of the expression of the ADAMTSL2 gene.

### Validation of ADAMTSL2 gene expression in CRC cell lines

The CRC cell lines SW620 and LoVo were cultured in RPMI-1640 cell culture medium supplemented with 10% fetal bovine serum, while normal colorectal cells (FHC) were cultured in DMEM/F12 supplemented with 10% fetal bovine serum, 10 ng/ml cholera toxin, 0.005 mg/ml insulin, 0.005 mg/ml transferrin and 100 ng/ml hydrocortisone. Cells were grown in a humidified incubator at 37˚C with 5% $CO_2$. We use qRT-PCR to detect ADAMTSL2 expression in the above cell lines. The primer sequences used in our investigation were as delineated below:

β-actin-F1: 5′-CGGATGTCCACGTCACACTT-3′,
β-actin-R1: 5′-GTTGCTATCCAGGCTGTGCT-3′;
ADAMTSL2-F1: 5′-CGTGCAGTGGAAGCTGTG-3′,
ADAMTSL2-R1: 5′-CAGGTCTTGGTACACTCGGA-3′ [32].

### Statistics analysis

Statistical analysis was performed using the R software. It is important to note that a p-value of less than 0.05 was considered to indicate statistical significance in this analysis.

## Results

### ADAMTSL2 is aberrantly expressed in pan-cancer and CRC

As shown in **S1 Table**, 644 patients are in this study. ADAMTSL2 was aberrantly expressed in 14 tumors (**Fig 1A**). ADAMTSL2 expression was higher in CRC compared to normal tissue ($2.670 \pm 0.040$ vs. $1.678 \pm 0.063$, $p < 0.001$) (**Fig 1B**). ADAMTSL2 expression was higher in CRC tissues compared to paired normal tissues ($2.788 \pm 0.165$ vs. $1.665 \pm 0.062$, $p < 0.001$) (**Fig 1C**). The area under the curve (AUC) of ADAMTSL2 was determined to be 0.815 (0.774–0.857) (**Fig 1D**).

### Association of ADAMTSL2 with clinical characteristics in patients with CRC

ADAMTSL2 expression was found to be significantly associated with the pathologic N stage ($p < 0.001$), pathologic stage ($p < 0.001$), age ($p < 0.001$), histological type ($p < 0.001$), and neoplasm type ($p = 0.001$) in patients with CRC (**S1**, **S2 Tables** and **Fig 2**).

### Association of ADAMTSL2 with prognosis in patients with CRC

ADAMTSL2 expression was found to be significantly associated with poor OS (HR: 1.67; 95% CI: 1.18–2.38; $p = 0.004$), PFS (HR: 1.55; 95% CI: 1.14–2.11; $p = 0.005$) and DSS (HR: 1.83;

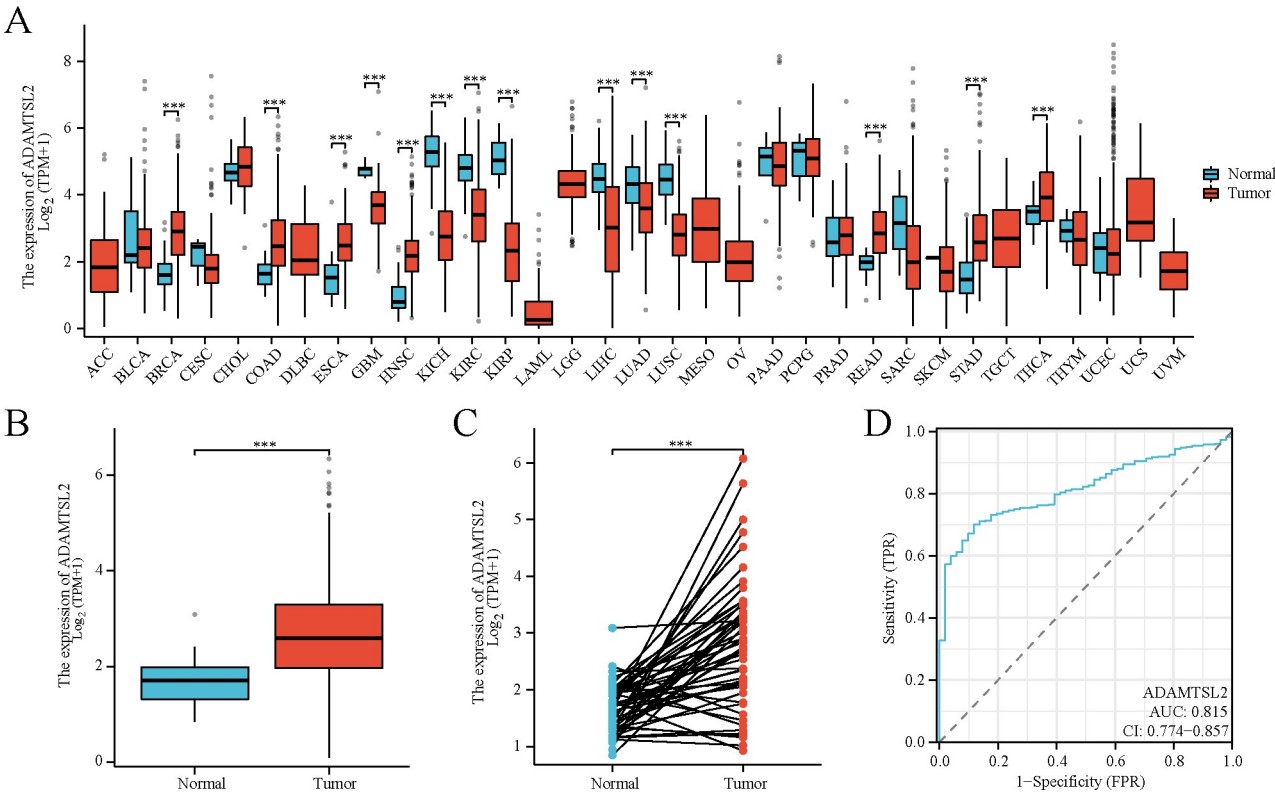

**Fig 1. ADAMTSL2 is aberrantly expressed in pan cancer and CRC.** (A) Difference expression of ADAMTSL2 in pan cancer and normal tissues. (B) Difference expression of ADAMTSL2 in CRC and normal colorectal tissues. (C) Difference expression of ADAMTSL2 in CRC and paired normal colorectal tissues. (D) Effectiveness of ADAMTSL2 expression in distinguishing CRC tissues from non-tumor tissues (ROC curve). ns, $p \geq 0.05$; *, $p < 0.05$; **, $p < 0.01$; ***, $p < 0.001$.

95% CI: 1.16–2.89; $p = 0.010$) in CRC patients (**Fig 3**). The results of the univariate analysis showed that the expression of ADAMTSL2 (HR: 1.672; 95% CI: 1.176–2.381; $p = 0.004$), T stage (HR: 2.468; 95% CI: 1.327–4.589; $p = 0.004$), N stage (HR: 2.627; 95% CI: 1.831–3.769; $p < 0.001$), pathologic stage (HR: 2.988; 95% CI: 2.042–4.372; $p < 0.001$), and age (HR: 1.939; 95% CI: 1.320–2.849; $p < 0.001$) were significantly associated with OS (**S3 Table**). The results of the multivariate analysis showed that the expression of ADAMTSL2 (HR: 1.654; 95% CI: 1.135–2.410; $p = 0.009$), N stage (HR: 0.273; 95% CI: 0.108–0.690; $p = 0.006$), pathologic stage (HR: 9.740; 95% CI: 3.754–25.272; $p < 0.001$), and age (HR: 2.749; 95% CI: 1.810–4.175; $p < 0.001$) were significantly associated with OS (**S3 Table** **and Fig 4A**). The data mentioned above indicate that ADAMTSL2 serves as an autonomous prognostic determinant for patients with CRC. We combine ADAMTSL2 expression levels and clinical features to construct a nomogram to predict the probability of survival at 1, 3, and 5 years in CRC patients (**Fig 4B**).

## ADAMTSL2-related pathways

We identified 45 sets of genes significantly differentially enriched in the expression phenotype of ADAMTSL2 by GSEA analysis. The analysis of the datasets revealed that extracellular matrix receptor (ECM-receptor) interaction, transforming growth factor β (TGF-β) signaling pathway, and more, were among the top 9 datasets with low P values (**Fig 5**).

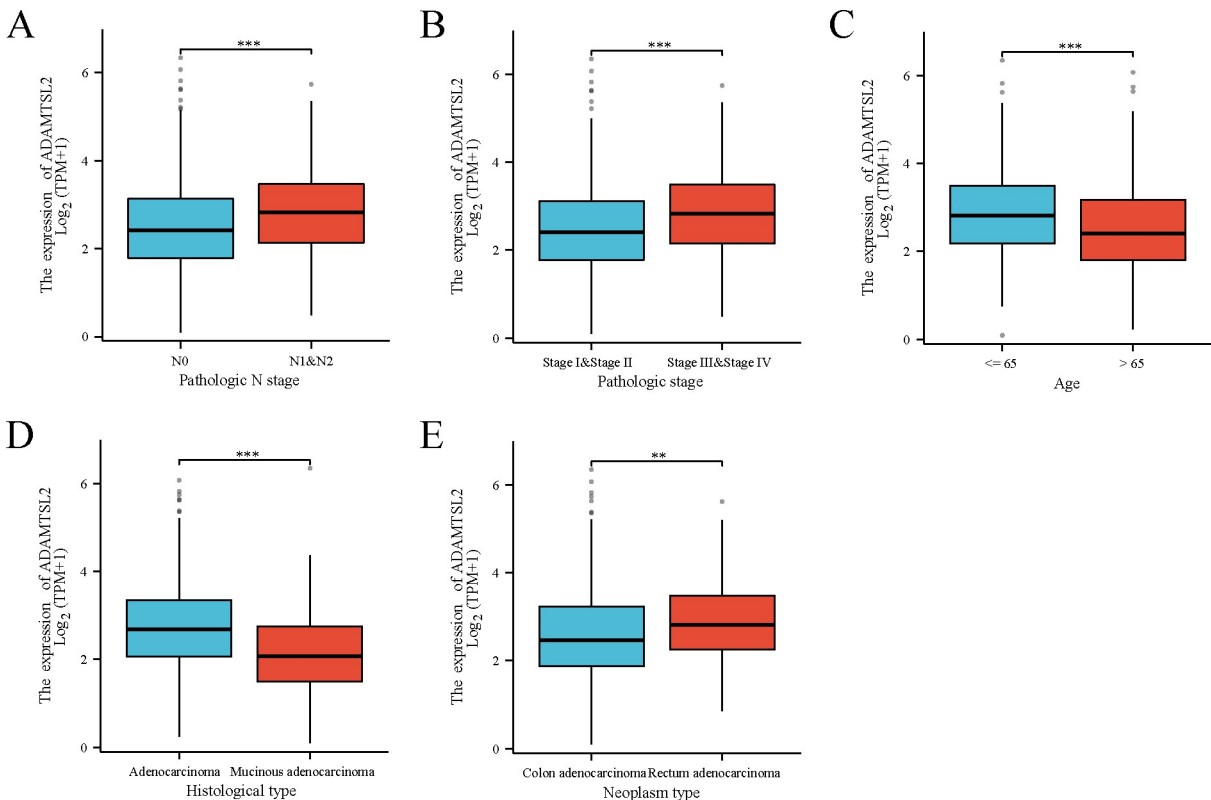

**Fig 2. Correlation of ADAMTSL2 expression with clinical characteristics in CRC.** (A) N stage. (B) Pathological stage. (C) Age. (D) Histological type. (E) Neoplasm type. Significance markers: **, $p < 0.01$; ***, $p < 0.001$.

## ADAMTSL2 were associated with immune infiltration and immune checkpoint genes

ADAMTSL2 expression was negatively correlated with the level of aDC ($p = 0.001$), cytotoxic cells ($p = 0.003$), T cells ($p < 0.001$), T helper cells ($p < 0.001$), Th1 cells ($p = 0.030$), and Th2 cells ($p < 0.001$), and positively correlated with that of Eosinophils ($p = 0.001$), iDC ($p < 0.001$), Mast cells ($p = 0.001$), NK CD56dim cells ($p = 0.001$), NK cells ($p < 0.001$), pDC

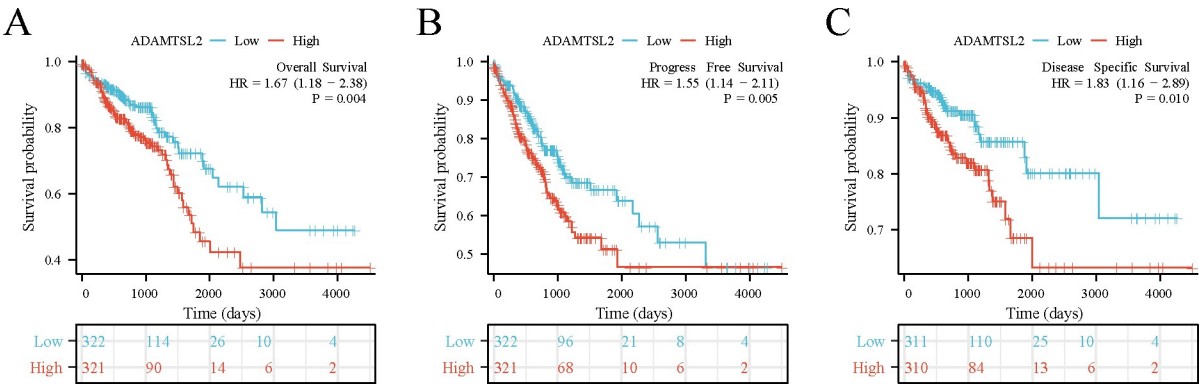

**Fig 3. ADAMTSL2 expression in CRC is associated with a poor prognosis in patients with CRC.** (A) OS. (B) PFS. (C) DSS.

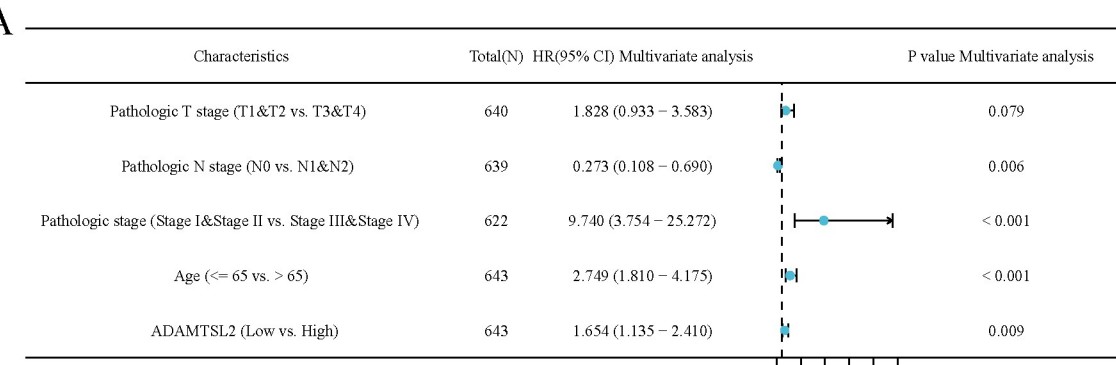

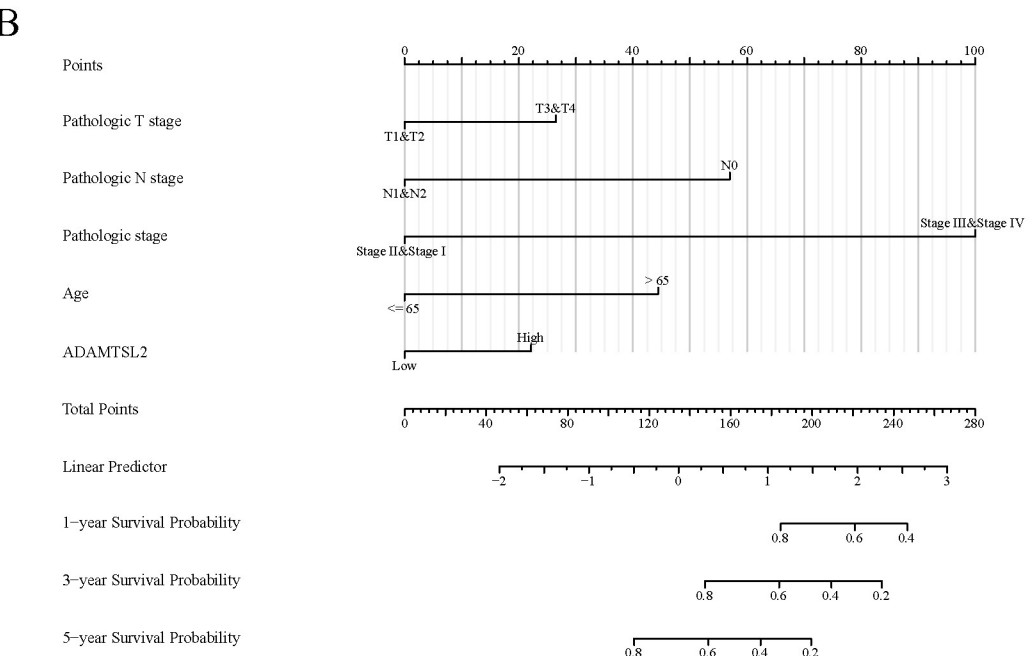

**Fig 4. ADAMTSL2 was an independent variable for predicting OS in CRC.** (A) Forest plot display of the results of the multifactorial Cox regression analysis of ADAMTSL2 and clinical characteristics in CRC. (B) Nomograms were constructed to predict the probability of OS at 1-, 3-, and 5- years in CRC.

($p < 0.001$), TFH ($p < 0.001$), and TReg ($p = 0.002$) (**Fig 6A**). The high expression ADAMTSL2 group had a significantly higher stromal score ($p < 0.001$) compared to the low expression ADAMTSL2 group (**Fig 6B**).

In READ patients, ADAMTSL2 expression was significantly and positively correlated with PDCD1 and SIGLEC15 expression (**Fig 7**). In patients with COAD, a positive correlation was observed between the expression of ADAMTSL2 and SIGLEC15, while a negative correlation was found with the expression of CD274, HAVCR2, LAG3 and TIGIT (**Fig 7**). The results suggested that ADAMTSL2 was associated with several immune checkpoint genes in CRC.

## Correlation between the expression of ADAMTSL2 and TMB/MSI

In CRC, the expression of ADAMTSL2 showed a significant negative correlation with TMB ($p = 6.4 \, e^{-10}$, **Fig 8A**). In CRC, the expression of ADAMTSL2 showed a significant negative correlation with MSI ($p = 2.41 \, e^{-07}$, **Fig 8B**).

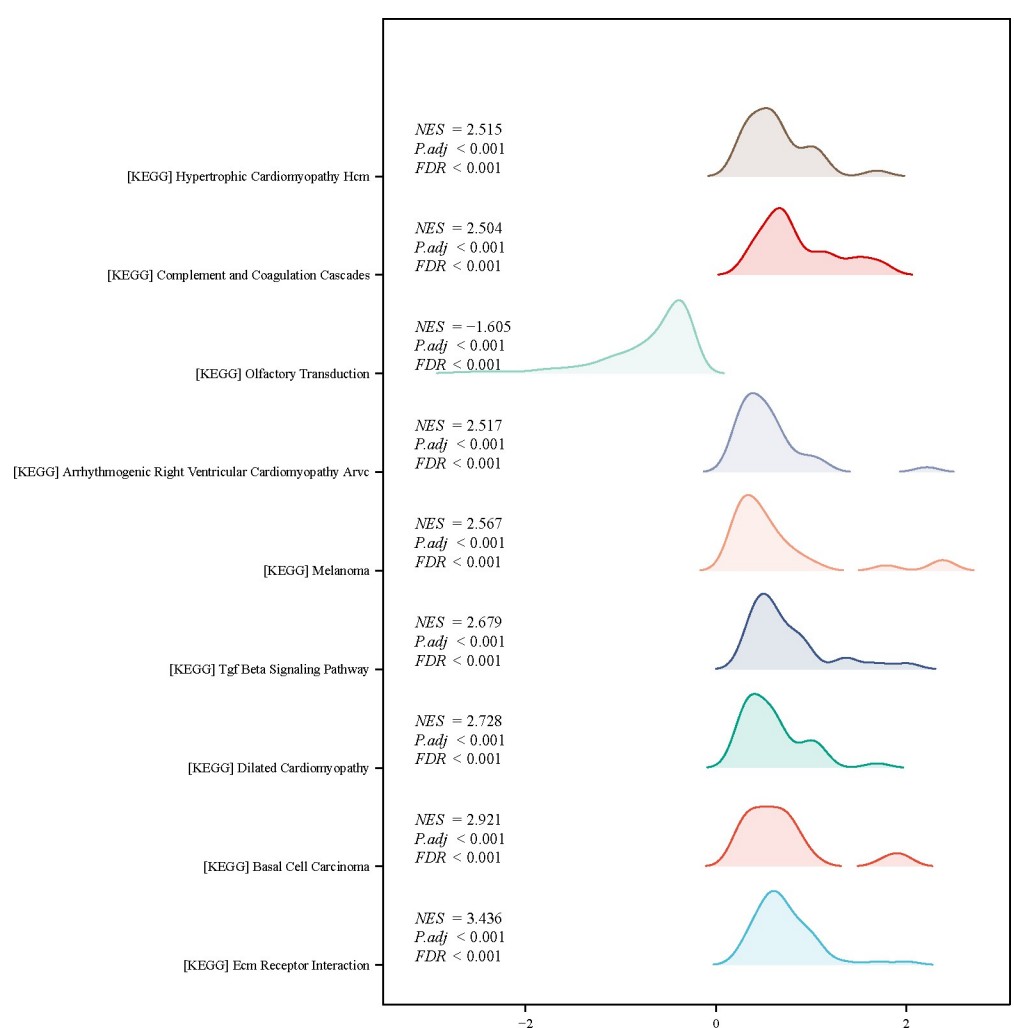

**Fig 5. Enrichment analysis of ADAMTSL2 in CRC (GSEA).**

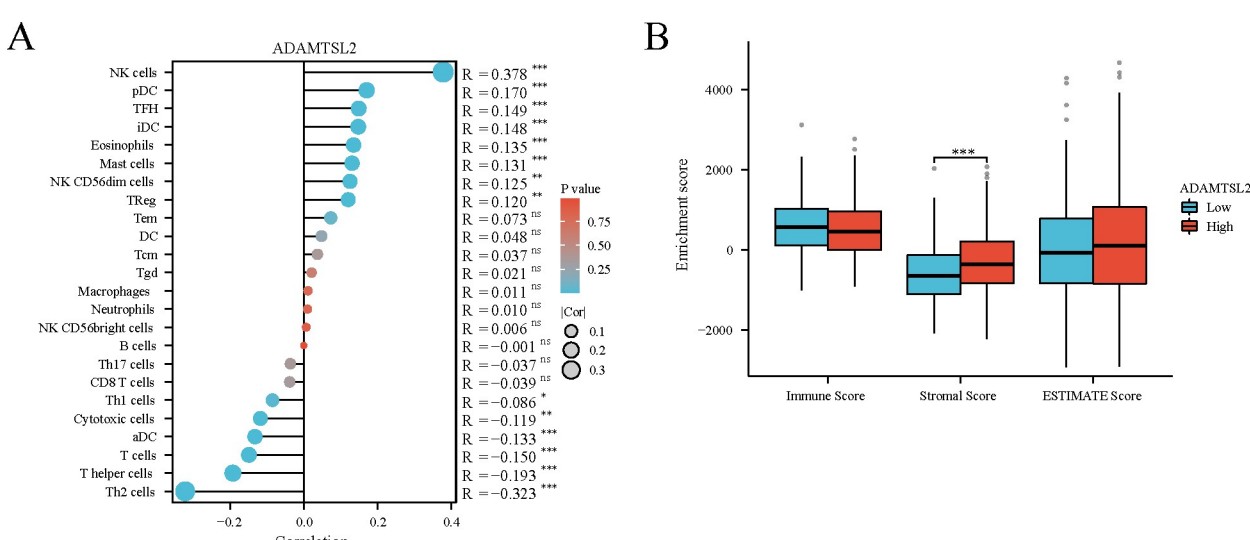

**Fig 6. ADAMTSL2 expression was associated with immune infiltration and immune score in CRC.** (A) Lollipop plot. (B) Grouped comparison plots. ns, $p \geq 0.05$; *, $p < 0.05$; **, $p < 0.01$; ***, $p < 0.001$.

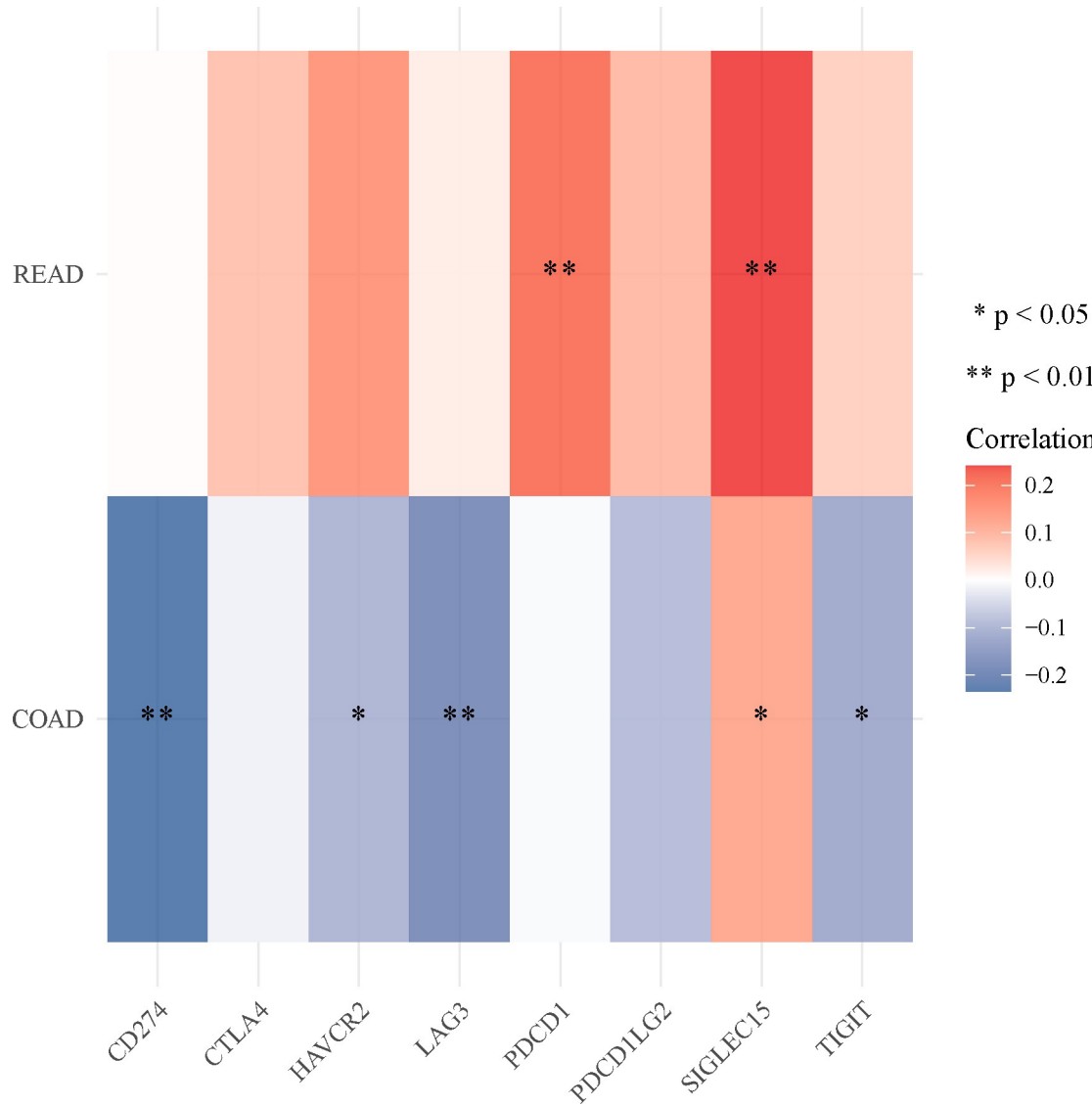

**Fig 7. The expression of ADAMTSL2 in CRC was associated with immune checkpoint genes.**

### Correlation between the expression of ADAMTSL2 and mRNAsi

Cancer progression involves progressive loss of differentiated phenotype and the acquisition of progenitor / stem cell-like features. The expression of ADAMTSL2 in CRC showed a negative correlation with mRNAsi (**Fig 9**).

### ADAMTSL2 expression in single CRC cells was correlated with immune infiltration

As shown **in Fig 10**, ADAMTSL2 was up-regulated in multiple individual CRC cells, including CD8Tex, endothelial, fibroblasts, and myofibroblasts.

### ADAMTSL2 expression was correlated with drug sensitivity

The present study used the RNAactDrug database to investigate the potential link between ADAMTSL2 expression and drug sensitivity. Our findings indicated a positive correlation

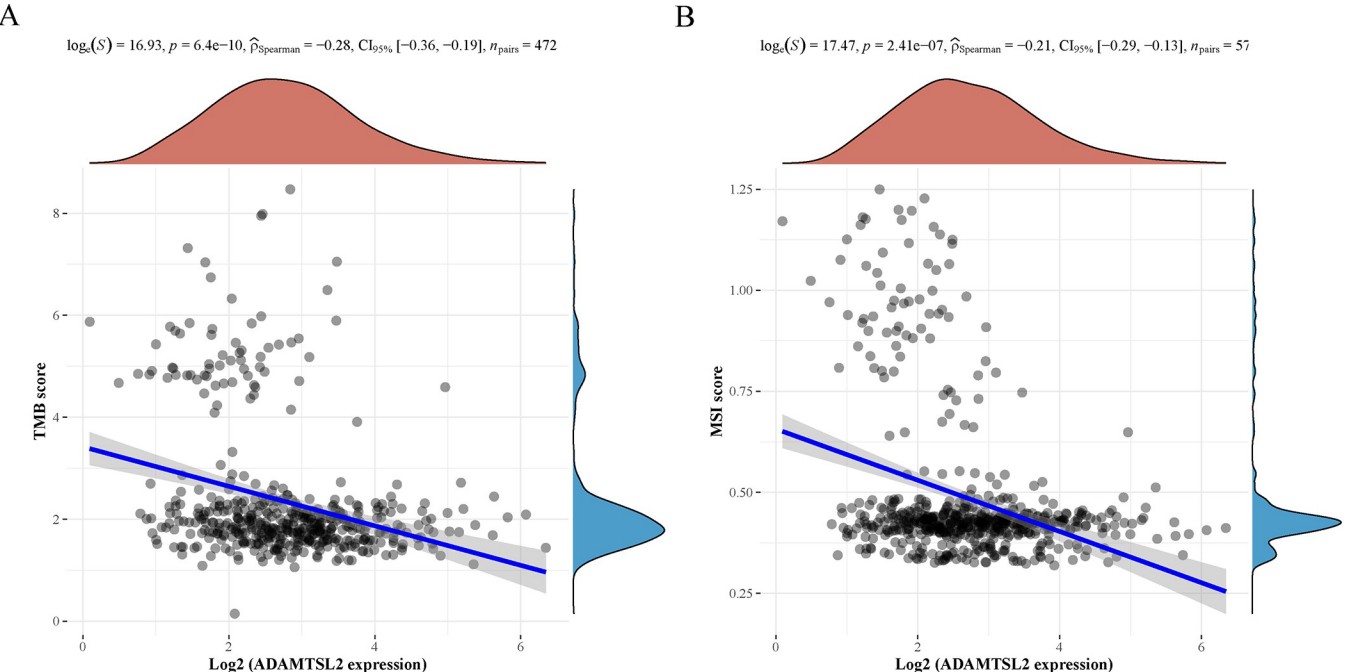

**Fig 8. ADAMTSL2 expression in CRC was associated with TMB/MSI.** (A) TMB. (B) MSI. Horizontal coordinates represent the gene and TMB/MSI correlation coefficients. Vertical coordinates on the graph correspond to various tumors, while the dot sizes represent the magnitude of the correlation coefficients. Furthermore, the colors employed in the schematic indicate the significance of the $p$-values, with bluer shades indicating smaller $p$-values.

between ADAMTSL2 expression and the sensitivity of [3-keto-bmt(sup 1)]-[val(sup 2)]-cyclosporin, 1,4-pentadien-3-one, 1,5-di-3-pyridyl-, navan, benzonitrile, 4-(3-oxo-3-phenyl-1-propenyl)-, and dihydroartemisinyl ether, stereoisomer of nsc-685988 (**Table 1**). On the contrary, the study revealed a negative correlation between ADAMTSL2 expression and the sensitivity of T0901317, Vorinostat, XMD14-99, 9-nitro-10-methoxy-20(s)-camptothecin and thalphenine chloride (**Table 1**). These results suggested a possible association between ZP3 and drug resistance to some drugs.

## Somatic variants of ADAMTSL2 in CRC

As shown in **Fig 11A**, the somatic variants of ADAMTSL2 included 10 missense mutations, 1 nonsense mutation, 3 frame shift del, 1 frame shift ins, and 1 splice site. The somatic mutation rate of ADAMTSL2 in CRC was 3.61%. As shown in **Fig 11B**, the top 10 mutated genes in the high expression group ADAMTSL2 and the low expression group included APC (77%), TP53 (59%), TTN (52%), KRAS (41%), SYNE1 (29%), MUC16 (29%), PIK3CA (25%), FAT4 (25%), RYR2(22%) and OBSCN (20%). As shown in **Fig 11C**, missense mutations were found to be the most prevalent variant classification, with SNPs being the most common, particularly the C > T mutation.

## ADAMTSL2 was aberrantly expressed in CRC tissues and cell lines

As shown in **Fig 12**, ADAMTSL2 is highly expressed in CRC tissues ($p < 0.001$) compared to normal colorectal tissues. As shown in **Fig 13**, ADAMTSL2 expression increased significantly in LoVo compared to FHC (2.713 ± 0.304 vs. 0.872 ± 0.209), ADAMTSL2 expression increased significantly in SW620 compared to FHC (1.786 ± 0.305 vs. 0.872 ± 0.209). Consequently, it can be concluded that ADAMTSL2 exhibited a significant up-regulation in CRC cell lines.

$$\log_e(S) = 17.83, p = 6.71e-25, \widehat{\rho}_{\text{Spearman}} = -0.40, \text{CI}_{95\%} [-0.46, -0.33], n_{\text{pairs}} = 62$$

**Fig 9. ADAMTSL2 expression in CRC was associated with mRNAsi.** The figure depicts the distribution of gene expression on the horizontal axis and the distribution of mRNAsi on the vertical axis; the density curve on the right side represents the trend of the mRNAsi distribution; the density curve on the upper side is the trend of the gene expression distribution; the uppermost values represent the correlation *p*-value, the correlation coefficient, and the correlation calculation method.

## Discussion

The occurrence and development of CRC is associated with the differential expression of a number of genes. Overexpression of Eva-1 homolog B (EVA1B) was associated with a poor prognosis in CRC patients [33]. C-X3-C motif chemokine receptor 1 (CX3CR1) may serve as a prognostic biomarker for CRC [34]. Down-regulation of TES (testin LIM domain protein) may portend a poor prognosis for CRC patients [35]. Centrosomal protein 55 (CEP55) showed promise as a potential diagnostic biomarker for CRC patients [36]. The high expression of the V-Set Immunoregulatory Receptor (VSIR) was associated with a favorable prognosis in CRC patients [37]. The extraction of novel molecular markers associated with the prognosis of CRC is necessary for the diagnosis and treatment of CRC.

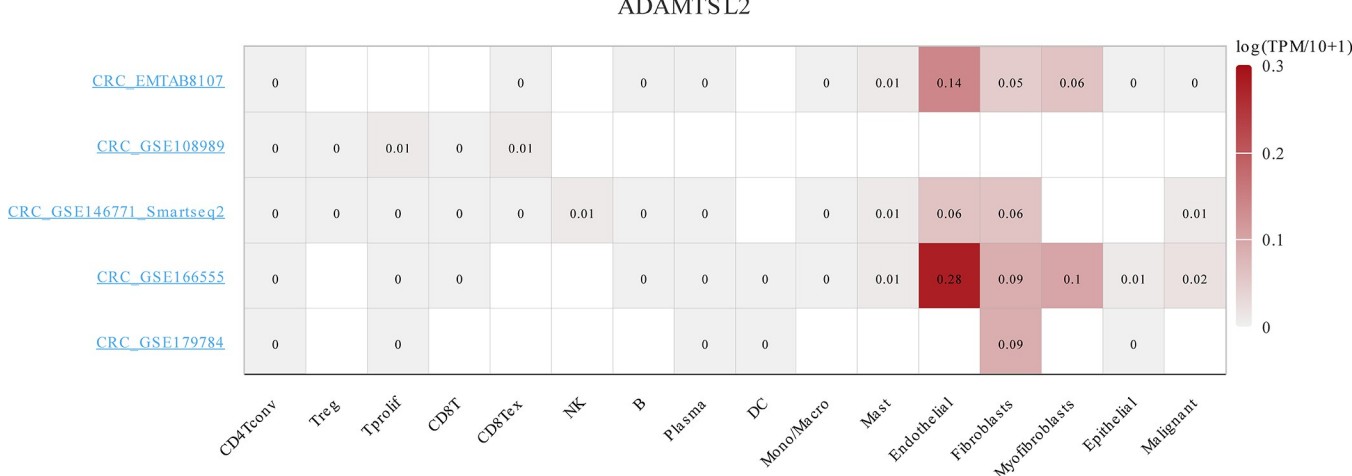

**Fig 10. ADAMTSL2 expression was related to immune infiltration in single CRC cells.**

In this study, the expression of ADAMTSL2 was higher in CRC tissues compared to normal colorectal tissues ($p < 0.001$). The high expression of ADAMTSL2 was associated with N stage ($p < 0.001$), pathologic stage ($p < 0.001$), age ($p < 0.001$), histological type ($p < 0.001$), and neoplasm type ($p = 0.001$). In patients with CRC, ADAMTSL2 expression was higher in patients with N1 & N2 stage compared to patients with N0 stage. In patients with CRC, ADAMTSL2 expression was higher in patients with stages III & IV compared to patients with stages I & II. These suggested that ADAMTSL2 may be involved in tumorigenesis and progression. ADAMTSL2 was correlated with poor OS ($p = 0.004$), PFS ($p = 0.005$) and DSS ($p = 0.010$) in patients with CRC. The expression of ADAMTSL2 ($p = 0.009$) has been identified as an autonomous prognostic determinant in patients with CRC.

ADAMTSL2 was found to improve myoblast differentiation by augmenting WNT signaling [38]. Furthermore, up-regulation of the ECM glycoprotein ADAMTSL2 was observed in heart failure, which was found to inhibit TGF-β signaling in cardiac fibroblasts [39]. This investigation has revealed an association between ADAMTSL2 and various pathways, including the ECM-receptor interaction, TGF-β signaling pathway, and more. The interactions between ECM and cellular receptors constitute one of the crucial pathways involved in the progression and metastasis of CRC [40]. Curcumin may inhibit LG5 (+) CRC by inducing autophagy and inhibiting the carcinogenic TFAP2 mediated ECM pathway [41]. TGF-β signaling is important

**Table 1. Drug sensitivity analysis of ADAMTSL2.**

| Compound | Source | Spearman.stat | Spearman.fdr | p value |
|---|---|---|---|---|
| [3-keto-bmt(sup 1)]-[val(sup 2)]-cyclosporin | CellMiner | 0.418 | 0.022 | 0.001 |
| 1,4-pentadien-3-one, 1,5-di-3-pyridyl- | CellMiner | 0.412 | 0.021 | 0.001 |
| navan | CellMiner | 0.407 | 0.029 | 0.001 |
| benzonitrile, 4-(3-oxo-3-phenyl-1-propenyl)- | CellMiner | 0.407 | 0.032 | 0.001 |
| dihydroartemisinyl ether, stereoisomer of nsc-685988 | CellMiner | 0.400 | 0.037 | 0.002 |
| T0901317 | GDSC | -0.174 | 0.000 | 0.000 |
| Vorinostat | GDSC | -0.175 | 0.000 | 0.000 |
| XMD14-99 | GDSC | -0.177 | 0.000 | 0.000 |
| 9-nitro-10-methoxy-20(s)-camptothecin | CellMiner | -0.452 | 0.037 | 0.000 |
| thalphenine chloride | CellMiner | -0.460 | 0.016 | 0.000 |

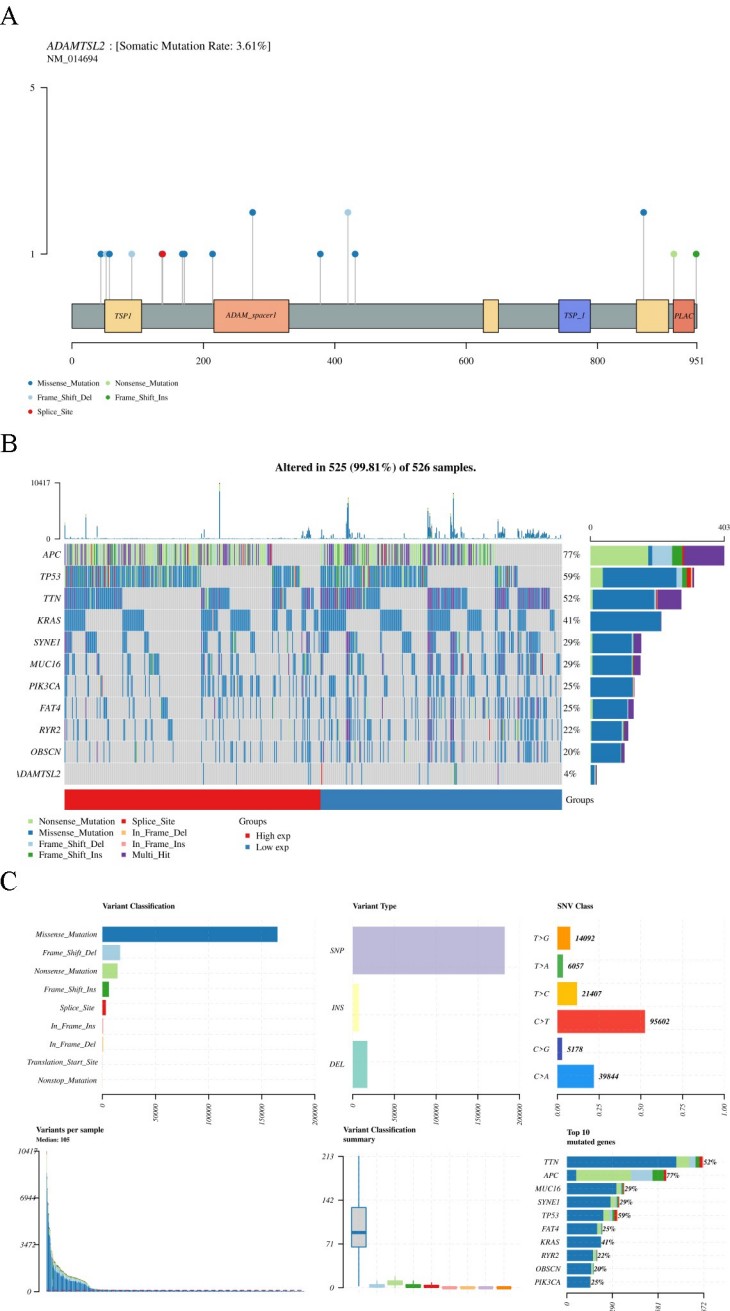

**Fig 11. Somatic mutations of ADAMTSL2 in CRC.** (A) Lollipop plot showing the distribution of mutations of the ADAMTSL2 gene. (B) Oncoplot showing the somatic landscape of ADAMTSL2 in the CRC cohort. Genes are sorted by mutation frequency and samples are sorted by disease histology, as indicated in the annotation column (bottom). The sidebar plot shows the transformed log10 Q values estimated by MutSigCV. Mutation information for each gene in each sample is shown in the waterfall plot, where the different colors with specific annotations at the bottom indicate the various types of mutations. The vignettes above the legend show the number of mutation burdens. (C) Cluster summary plots show the distribution of variants according to variant classification, type, and SNV category. The bottom (from left to right) indicates the mutational load for each sample (variant classification type). The stacked bar graph shows the top ten mutated genes.

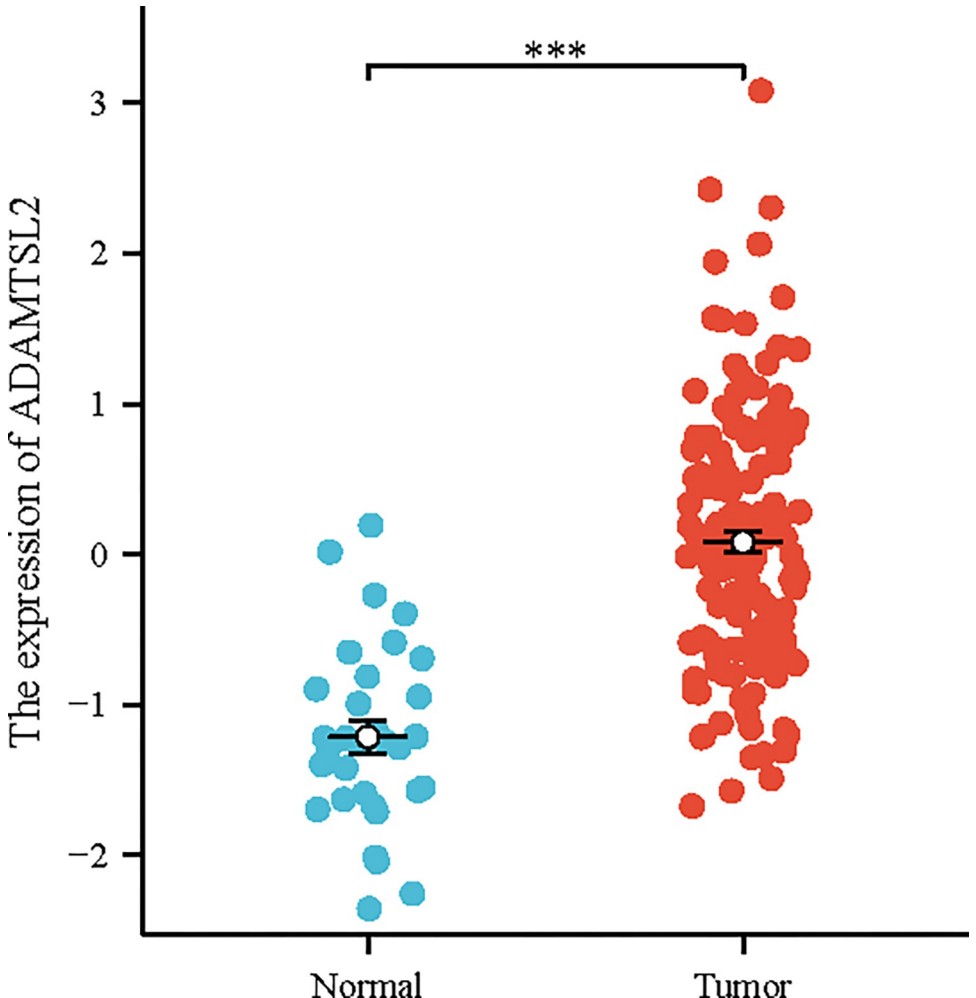

**Fig 12. ADAMTSL2 was aberrantly expressed in the CRC of GSE71187.**

in the context of inflammation and tumorigenesis by modulating cell growth, differentiation, apoptosis, and homeostasis [42]. The specific mechanism by which ADAMTSL2 mediates the occurrence of CRC through these pathways needs further investigation.

Examining immune infiltration in CRC has emerged as a prominent area of research [43]. Immunotherapy has achieved strong antitumor efficacy in many cancers [44]. Other immuno-therapies for CRC patients are still being developed, despite the good efficacy of immune checkpoint inhibitors (ICI) [45]. This study aimed to investigate the relationship between ADAMTSL2 expression and immunity in CRC. The expression of ADAMTSL2 in CRC was found to be correlated with the infiltration of various types of immune cells, including aDC, cytotoxic cells, T cells, T helper cells, Th1 cells, Th2 cells, Eosinophils, iDC, Mast cells, NK CD56dim cells, NK cells, pDC, TFH, and TReg. These observed associations may imply potential mechanisms by which ADAMTSL2 hinders the functioning of aDC, cytotoxic cells, T cells, T helper cells, Th1 cells, and Th2 cells, while improving the function of Eosinophils, iDC, Mast cells, NK CD56dim cells, NK cells, pDC, TFH, and TReg. ADAMTSL2 expression in patients with CRC is associated with CD274, HAVCR2, LAG3, PDCD1, SIGLEC15 and TIGIT. There is a significant positive correlation between the expression level of ADAMTSL2 and the

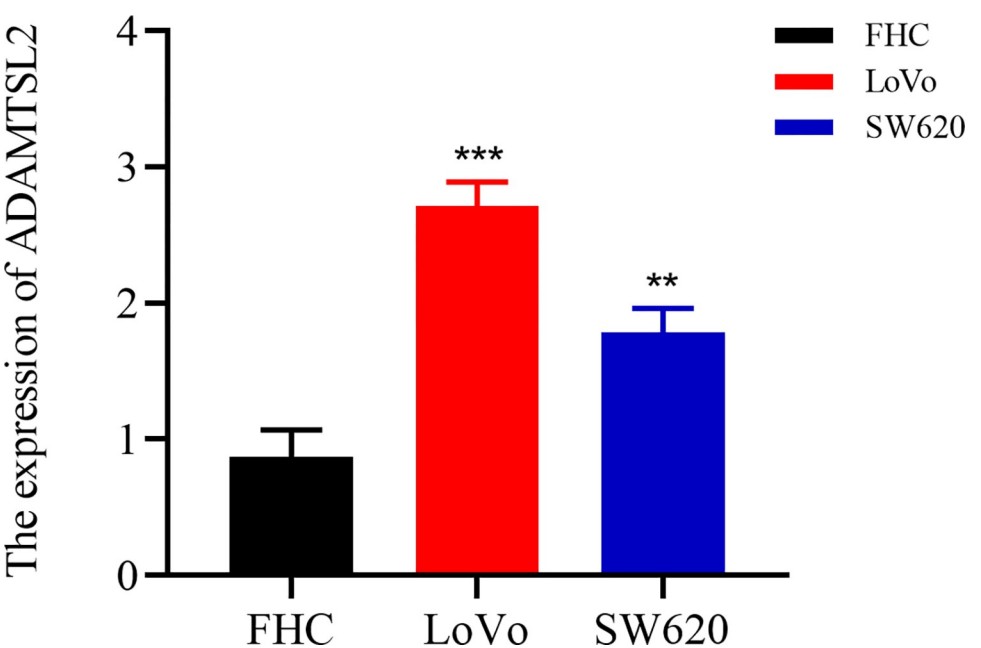

**Fig 13. ADAMTSL2 is aberrantly expressed in CRC cell lines.**

stromal score. This indicates that ADAMTSL2 can affect the CRC microenvironment by altering stromal cells. The specific mechanism between ADAMTSL2 and immune infiltration and immune checkpoints in CRC needs to be further investigated.

More than 80% of MSI high tumors have a TMB value of >20 mutations/Mb [27]. Furthermore, the study examined the correlation between ADAMTSL2 expression and MSI or TMB. The results revealed a significant association between ADAMTSL2 expression and MSI and TMB in CRC. ADAMTSL2 in CRC is negatively correlated with TMB and MSI, indicating that ADAMTSL2 may reflect the cancer immunogenicity of CRC. The influence of ADAMTSL2 expression on the response of CRC patients to immune checkpoint therapy suggested the potential to use immunotherapy in CRC treatment guided by ADAMTSL2.

Cancer stem cells (CSCs) propose that CSCs are central to carcinogenesis [46]. mRNAsi is an index calculated based on expression data [47]. In this study, we found that ADAMTSL2 expression is negatively correlated with mRNAsi in CRC. The specific mechanism between ADAMTSL2 and CSC in CRC needs further study.

The relationship between ADAMTSL2 expression and drug sensitivity remains uncertain. In this study, we used the analysis of the RNAactDrug database to find that ADAMTSL2 is associated with several sensitivities, including [3-keto-bmt(sup 1)]-[val(sup 2)]-cyclosporin, 1,4-pentadien-3-one, 1,5-di-3-pyridyl-, navan, benzonitrile, 4-(3-oxo-3-phenyl-1-propenyl)-, and dihydroartemisinyl ether, stereoisomer of nsc-685988. This upregulation of ADAMTSL2 can potentially contribute to drug therapy and could be associated with the development of drug resistance.

However, there are some limitations to this study. First, this study is a clinical significance and regulatory network analysis of ADAMTSL2 in CRC using public databases TCGA and GEO and these results need to be further validated in real-world samples. Second, the molecular mechanism of ADAMTSL2-mediated CRC development needs to be further experimentally validated.

## Conclusions

ADAMTSL2 expression was significantly increased in CRC, suggesting a poorer OS. ADAMTSL2 may play a role in the development of CRC through various pathways, including ECM-receptor interaction, TGF-β signaling pathway, and more. ADAMTSL2 expression in CRC was significantly associated with immune infiltration, immune checkpoint genes, TMB/MSI, and mRNAsi in CRC. ADAMTSL2 expression was significantly and negatively correlated with [3-keto-bmt(sup 1)]-[val(sup 2)]-cyclosporin, 1,4-pentadien-3-one, 1,5-di-3-pyridyl-, navan, benzonitrile, 4-(3-oxo-3-phenyl-1-propenyl)-, and dihydroartemisinyl ether, stereoisomer of nsc-685988 in CRC. This finding suggested that ADAMTSL2 may serve as a potential biomarker for predicting the prognosis and guiding immunotherapy in CRC.

## Supporting information

**S1 Table. Correlation of ADAMTSL2 expression with clinical characteristics in CRC.** (DOCX)

**S2 Table. Correlation between ADAMTSL2 expression and clinical characteristics in CRC (logistic analysis).** (DOCX)

**S3 Table. Univariate and multivariate analysis of DSS and clinical characteristics in CRC (Cox regression).** (DOCX)

## Acknowledgments

The authors thank the TCGA and GEO databases for providing the data.

## Author Contributions

**Conceptualization:** Zhe Huang, Pengzhu Cai.

**Data curation:** Jiaming Qi, Jinglin Pang.

**Formal analysis:** Kang Huang, Huagui Li.

**Funding acquisition:** Zhe Huang.

**Validation:** Xu Hu, Yiqiu Wei, Yousheng Lai.

**Writing – original draft:** Zhe Huang, Pengzhu Cai.

**Writing – review & editing:** Zhe Huang, Xu Hu, Yiqiu Wei, Yousheng Lai, Jiaming Qi, Jinglin Pang, Kang Huang, Huagui Li, Pengzhu Cai.

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
