## [Decision Letter · Decision Letter 0]

30 Nov 2023

PONE-D-23-31014Identification of ADAMTSL2 as a potential prognosis biomarker associated with immune infiltrates and drug sensitivity in colorectal cancerPLOS ONE

Dear Dr. Huang,

Thank you for submitting your manuscript to PLOS ONE. After careful consideration, we feel that it has merit but does not fully meet PLOS ONE’s publication criteria as it currently stands. Therefore, we invite you to submit a revised version of the manuscript that addresses the points raised during the review process, especially in the results section.

We look forward to receiving your revised manuscript.

Kind regards,

Shiki Fujino, M.D.

Academic Editor

PLOS ONE

Journal Requirements:

2. a.For studies reporting research involving human participants, PLOS ONE requires authors to confirm that this specific study was reviewed and approved by an institutional review board (ethics committee) before the study began. Please provide the specific name of the ethics committee/IRB that approved your study, or explain why you did not seek approval in this case.

b. Please provide additional details regarding participant consent. In the ethics statement in the Methods and online submission information, please ensure that you have specified what type you obtained (for instance, written or verbal, and if verbal, how it was documented and witnessed). If your study included minors, state whether you obtained consent from parents or guardians. If the need for consent was waived by the ethics committee, please include this information.

https://jcancer.org/ms/doc/2019/epub/61415j3/61415j3.pdf

In your revision ensure you cite all your sources (including your own works), and quote or rephrase any duplicated text outside the methods section. Further consideration is dependent on these concerns being addressed.

"This study was supported by Guangdong Medical Research Foundation Project in 2020 (No. B2020112), Competitive Allocation Project of Special Funds for Science and Technology Development by Zhanjiang City in 2021 (No. 2021A05080), Clinical Research Project Funded by Affiliated Hospital of Guangdong Medical University in 2019 (No. LCYJ20198001)."

7. Your ethics statement should only appear in the Methods section of your manuscript. If your ethics statement is written in any section besides the Methods, please move it to the Methods section and delete it from any other section. Please ensure that your ethics statement is included in your manuscript, as the ethics statement entered into the online submission form will not be published alongside your manuscript. 

Reviewers' comments:

Reviewer's Responses to Questions

**Comments to the Author**

1. Is the manuscript technically sound, and do the data support the conclusions?

Reviewer #1: No

Reviewer #2: Yes

2. Has the statistical analysis been performed appropriately and rigorously? 

Reviewer #1: I Don't Know

Reviewer #2: N/A

3. Have the authors made all data underlying the findings in their manuscript fully available?

Reviewer #1: Yes

Reviewer #2: Yes

4. Is the manuscript presented in an intelligible fashion and written in standard English?

Reviewer #1: No

Reviewer #2: No

5. Review Comments to the Author

Reviewer #1: In the current manuscript the authors analyzed the prognostic roles of ADAMTSL2 in colorectal cancer.

below are my comment about this manuscript

1.The title of the manuscript is unclear. Please add detailed information about the patient group involved in the analysis. For example, include details about tumor stage and whether patients received any treatment before the analysis.

Abstract

1.The background is insufficient, and the study objective is not precisely described.

2.The study method is unclear; please provide a more detailed description.

3.The results are poorly written, and some are confusing, such as the statement, "ADAMTSL2 was associated with ECM-receptor interaction, basal cell carcinoma, dilated cardiomyopathy, TGF-beta signaling pathway, melanoma, arrhythmogenic right ventricular cardiomyopathy (ARVC), Olfactory transduction, complement and coagulation cascades, and hypertrophic cardiomyopathy (HCM)." Please summarize only the results directly related to the study objective.

4.The conclusion needs clarification.

Introduction and Method Section:

1.In section 2.5, subgroups were classified into two categories, 0-50 and 50-100. Please specify the criteria used to divide patients into these subgroups.

2.In section 2.7, the second paragraph lacks clarity and does not describe the method used to analyze the correlation between ADAMTSL2 and immune-related genes. Additionally, in section 2.8, the analysis method used to examine the correlation of ADAMTSL2 with TMB/MSI is not described.

3.Provide more explanation about the analysis used in section 2.9.

4.Clarify section 2.10 in more detail.

Results:

1.In section 3.1, the heading does not describe the findings clearly. The phrase "The results suggested that ADAMTSL2 was associated with tumorigenesis" needs clarification based on the data supporting this conclusion. Another conclusion, "indicating that ADAMTSL2 could be a new biomarker to differentiate CRC from non-tumor tissues," lacks sufficient data for such a claim.

2.In section 3.2, the description of the results in Figure 2 is not entirely correct and lacks clarity. Please rewrite this section.

3.In section 3.3, the results description is unclear and not entirely accurate. Additionally, there is a discrepancy between your above reported results that ADAMTSL2 is significantly low-expressed in CRC compared with normal colorectal tissue and in this section you stating that patients with high expression of ADAMTSL2 have significantly poor OS and DSS. This contradiction needs clarification.

4.In section 3.4, concentrate the analysis on pathways related to CRC and ignore unrelated pathways.

5.In section 3.5, explain the meaning of READ and COAD patients.

6.Clarify the main results in Sections 3.6, 3.7, and 3.8.

7.Rewrite section 3.9 for better clarity.

8.The results in section 3.11 contradict those in section 3.1. The expression of ADAMTSL2 is reported as lower in CRC compared to normal tissue in section 3.1, while section 3.11 you reports that ADAMTSL2 is highly expressed in CRC tissues. This discrepancy is unacceptable and requires attention.

Reviewer #2: In this study, Dr. Huang and their colleagues conducted a comprehensive analysis to evaluate the potential prognostic significance of the ADAMTS-like-2 (ADAMTSL2), as well as the correlation between ADAMTSL2 expression and immune infiltrates as well as drug sensitivity in colorectal cancer. Furthermore, they validated the elevated expression of ADAMTSL2 in colorectal cancer cell lines compared to normal cell line by Real-Time PCR. While the effort and perspective of this paper is interesting, there are several concerns and major revisions are expected.

1. In the introduction section, authors stated “The members of ADAMTSL (a disintegrin-like and metalloproteinase domain with thrombospondin type 1 motif-like) family that secrete glycoproteins are similar to ADAMTS proteases in structure”. This family members secrete glycoproteins? Or this family is kind of secreted glycoproteins. Please rewrite this sentence clearly.

2. Which database or algorithm was used to evaluate the relationship between ADAMTSL2 and TMB/MSI?

3. Why did the authors include references for some parts of material method section that, in my opinion, it seems unnecessary (such as Ref [32]).

4. In the result section, 3.1. ADAMTSL2 is aberrantly expressed in pan cancer and CRC part, I think the authors have inaccurately written this paragraph. They stated “The expression of ADAMTSL2 were lower in CRC compared to normal tissue (2.670 ± 0.040 vs. 1.678 ± 0.063, p < 0.001) (Figure 1B). The expression of ADAMTSL2 were lower in CRC tissues compared to paired normal tissues” . But I think ADAMTSL2 expression is higher in CRC compared to normal tissue.

5. In the result section, 3.3. Association of ADAMTSL2 with survival in patients with CRC part, Table S3 shows the results of univariate and multivariate analyses, not the correlation between ADAMTSL2 expression and clinicopathologial characteristics. I think the authors have inaccurately written some parts of this paragraph (the results of univariate analysis).

6. Authors have stated that ADAMTSL2 is highly expressed in CRC tissues compared with normal colorectal tissues in “3.11. ADAMTSL2 was aberrantly expressed in CRC tissues and cell lines part”. However, they have not specified the characteristics of these tissues. Or which correlation was between ADAMTSL2 expression and clinicopathological features in these tissues.

7. The discussion part was inadequately written. The authors should interpret the results. For example, how this gene is related to the mentioned signaling pathways? Or how the authors interpret the relationship between ADAMTSL2 expression and TMB/MSI? Or the relationship of the ADAMTSL2 high expression group with higher stromal score?

8. In the present study, ADAMTSL2 expression is associated with different immune infiltration, some immune checkpoint genes, and signaling pathways. These are highly interesting results. One possible method would be to use siRNA to downregulate the ADAMTSL2 expression in the CRC cell lines or mouse model, assess the expression of some investigated genes, and confirm by experimental examination.

9. The figures presented have poor contrast, including Figures 1-8.

10. Abbreviations should be fully described the first time they appear, such as COAD, READ, IGLEC15, IDO1, CD274, HAVCR2, PDCD1, CTLA4, LAG3, and PDCD1LG2, …

11. There are some grammatical errors throughout, such as:

ADAMTSL2 were associated with

The occurrence and development of CRC is associated

The expression of ADAMTSL2 were lower in CRC tissues compared to paired normal tissues

6. PLOS authors have the option to publish the peer review history of their article (what does this mean?). If published, this will include your full peer review and any attached files.

Reviewer #1: No

Reviewer #2: No

---

## [Author Response · Author response to Decision Letter 0]

5 Feb 2024

Dear Editor:

We have carefully studied the valuable comments from reviewers and tried our best to revise the manuscript titled “Identification of ADAMTSL2 as a potential prognosis biomarker associated with immune infiltrates and drug sensitivity in colorectal cancer” (PONE-D-23-31014). The amendment has been marked with red. We have made detailed changes throughout the text. The point to point responds to the reviewer’s comments are listed in the revised text.

Reviewer #1: 

In the current manuscript the authors analyzed the prognostic roles of ADAMTSL2 in colorectal cancer.

below are my comment about this manuscript

1.The title of the manuscript is unclear. Please add detailed information about the patient group involved in the analysis. For example, include details about tumor stage and whether patients received any treatment before the analysis.

Response: Thanks for your comments. The title was improved according to your suggestion.

Abstract

1.The background is insufficient, and the study objective is not precisely described.

Response: Thanks for your comments. The background of the abstract was improved according to your suggestion.

2.The study method is unclear; please provide a more detailed description.

Response: Thanks for your comments. The abstract methods was improved according to your suggestion.

3.The results are poorly written, and some are confusing, such as the statement, "ADAMTSL2 was associated with ECM-receptor interaction, basal cell carcinoma, dilated cardiomyopathy, TGF-beta signaling pathway, melanoma, arrhythmogenic right ventricular cardiomyopathy (ARVC), Olfactory transduction, complement and coagulation cascades, and hypertrophic cardiomyopathy (HCM)." Please summarize only the results directly related to the study objective.

Response: Thanks for your comments. The abstract results were improved according to your suggestion.

4.The conclusion needs clarification.

Response: Thanks for your comments. The conclusions were improved according to your suggestion.

Introduction and Method Section:

1.In section 2.5, subgroups were classified into two categories, 0-50 and 50-100. Please specify the criteria used to divide patients into these subgroups.

Response: Thanks for your comments. Section 2.5 was improved according to your suggestion.

2.In section 2.7, the second paragraph lacks clarity and does not describe the method used to analyze the correlation between ADAMTSL2 and immune-related genes. Additionally, in section 2.8, the analysis method used to examine the correlation of ADAMTSL2 with TMB/MSI is not described.

Response: Thanks for your comments. Sections 2.7 and 2.8 were improved according to your suggestion.

3.Provide more explanation about the analysis used in section 2.9.

Response: Thanks for your comments. Section 2.9 was improved according to your suggestion.

4.Clarify section 2.10 in more detail.

Response: Thanks for your comments. Section 2.10 was improved according to your suggestion.

Results:

1.In section 3.1, the heading does not describe the findings clearly. The phrase "The results suggested that ADAMTSL2 was associated with tumorigenesis" needs clarification based on the data supporting this conclusion. Another conclusion, "indicating that ADAMTSL2 could be a new biomarker to differentiate CRC from non-tumor tissues," lacks sufficient data for such a claim.

Response: Thanks for your comments. The description was improved according to your suggestion.

2.In section 3.2, the description of the results in Figure 2 is not entirely correct and lacks clarity. Please rewrite this section.

Response: Thanks for your comments. Section 3.2 was improved according to your suggestion.

3.In section 3.3, the results description is unclear and not entirely accurate. Additionally, there is a discrepancy between your above reported results that ADAMTSL2 is significantly low-expressed in CRC compared with normal colorectal tissue and in this section you stating that patients with high expression of ADAMTSL2 have significantly poor OS and DSS. This contradiction needs clarification.

Response: Thanks for your comments. We checked the whole text carefully and Section 3.1 was corrected according to your suggestion.

4.In section 3.4, concentrate the analysis on pathways related to CRC and ignore unrelated pathways.

Response: Thanks for your comments. Section 3.4 was improved according to your suggestion.

5.In section 3.5, explain the meaning of READ and COAD patients.

Response: Thanks for your comments. The meaning of READ and COAD patients was supplemented in Section 2.1 according to your suggestion. 

6.Clarify the main results in Sections 3.6, 3.7, and 3.8.

Response: Thanks for your comments. Sections 3.6, 3.7, and 3.8 were improved according to your suggestion.

7.Rewrite section 3.9 for better clarity.

Response: Thanks for your comments. Section 3.9 was improved according to your suggestion.

8.The results in section 3.11 contradict those in section 3.1. The expression of ADAMTSL2 is reported as lower in CRC compared to normal tissue in section 3.1, while section 3.11 you reports that ADAMTSL2 is highly expressed in CRC tissues. This discrepancy is unacceptable and requires attention.

Response: Thanks for your comments. We checked the whole text carefully and the Section 3.1 was corrected according to your suggestion.

Reviewer #2: 

In this study, Dr. Huang and their colleagues conducted a comprehensive analysis to evaluate the potential prognostic significance of the ADAMTS-like-2 (ADAMTSL2), as well as the correlation between ADAMTSL2 expression and immune infiltrates as well as drug sensitivity in colorectal cancer. Furthermore, they validated the elevated expression of ADAMTSL2 in colorectal cancer cell lines compared to normal cell line by Real-Time PCR. While the effort and perspective of this paper is interesting, there are several concerns and major revisions are expected.

1. In the introduction section, authors stated “The members of ADAMTSL (a disintegrin-like and metalloproteinase domain with thrombospondin type 1 motif-like) family that secrete glycoproteins are similar to ADAMTS proteases in structure”. This family members secrete glycoproteins? Or this family is kind of secreted glycoproteins. Please rewrite this sentence clearly.

Response: Thanks for your comments. The information was improved according to your suggestion.

2. Which database or algorithm was used to evaluate the relationship between ADAMTSL2 and TMB/MSI?

Response: Thanks for your comments. The information was supplemented according to your suggestion.

3. Why did the authors include references for some parts of material method section that, in my opinion, it seems unnecessary (such as Ref [32]).

Response: Thanks for your comments. The references were improved according to your suggestion.

4. In the result section, 3.1. ADAMTSL2 is aberrantly expressed in pan cancer and CRC part, I think the authors have inaccurately written this paragraph. They stated “The expression of ADAMTSL2 were lower in CRC compared to normal tissue (2.670 ± 0.040 vs. 1.678 ± 0.063, p < 0.001) (Figure 1B). The expression of ADAMTSL2 were lower in CRC tissues compared to paired normal tissues” . But I think ADAMTSL2 expression is higher in CRC compared to normal tissue.

Response: Thanks for your comments. We checked the whole text carefully and the section 3.1 was corrected according to your suggestion.

5. In the result section, 3.3. Association of ADAMTSL2 with survival in patients with CRC part, Table S3 shows the results of univariate and multivariate analyses, not the correlation between ADAMTSL2 expression and clinicopathologial characteristics. I think the authors have inaccurately written some parts of this paragraph (the results of univariate analysis).

Response: Thanks for your comments. The description about the result of the univariate analysis was improved according to your suggestion.

6. Authors have stated that ADAMTSL2 is highly expressed in CRC tissues compared with normal colorectal tissues in “3.11. ADAMTSL2 was aberrantly expressed in CRC tissues and cell lines part”. However, they have not specified the characteristics of these tissues. Or which correlation was between ADAMTSL2 expression and clinicopathological features in these tissues.

Response: Thanks for your comments. We validated ADAMTSL2 expression in CRC cell lines, so we can’t present the correlation between ADAMTSL2 and clinical characteristics. We mention this limitation in the article. This study is a clinical significance and regulatory network analysis of ADAMTSL2 in CRC using the public databases of TCGA and GEO, and these results must be further validated in real world samples.

7. The discussion part was inadequately written. The authors should interpret the results. For example, how this gene is related to the mentioned signaling pathways? Or how the authors interpret the relationship between ADAMTSL2 expression and TMB/MSI? Or the relationship of the ADAMTSL2 high expression group with higher stromal score?

Response: Thanks for your comments. The Discussion section was improved according to your suggestion.

8. In the present study, ADAMTSL2 expression is associated with different immune infiltration, some immune checkpoint genes, and signaling pathways. These are highly interesting results. One possible method would be to use siRNA to downregulate the ADAMTSL2 expression in the CRC cell lines or mouse model, assess the expression of some investigated genes, and confirm by experimental examination.

Response: Thanks for your comments. We will actively apply for research funding based on existing research results. Once funding support is obtained, we will conduct more detailed research in the future.

9. The figures presented have poor contrast, including Figures 1-8.

Response: Thanks for your comments. Figure 1 was improved according to your suggestion.

10. Abbreviations should be fully described the first time they appear, such as COAD, READ, IGLEC15, IDO1, CD274, HAVCR2, PDCD1, CTLA4, LAG3, and PDCD1LG2, …

Response: Thanks for your comments. The abbreviations were improved according to your suggestion.

11. There are some grammatical errors throughout, such as:

ADAMTSL2 were associated with

The occurrence and development of CRC is associated

The expression of ADAMTSL2 were lower in CRC tissues compared to paired normal tissues

Response: Thanks for your comments. The whole text was improved according to your suggestion.

Thank you for considering our manuscript.

Yours sincerely,

Zhe Huang

---

## [Decision Letter · Decision Letter 1]

1 Apr 2024

PONE-D-23-31014R1ADAMTSL2 is associated with immune infiltration and predicts a poor prognosis in colorectal cancerPLOS ONE

Dear Dr. Huang,

Thank you for submitting your manuscript to PLOS ONE. After careful consideration, we feel that it has merit but does not fully meet PLOS ONE’s publication criteria as it currently stands. Therefore, we invite you to submit a revised version of the manuscript that addresses the points raised during the review process.

We look forward to receiving your revised manuscript.

Kind regards,

Shiki Fujino, M.D.

Academic Editor

PLOS ONE

Journal Requirements:

Reviewers' comments:

Reviewer's Responses to Questions

**Comments to the Author**

1. If the authors have adequately addressed your comments raised in a previous round of review and you feel that this manuscript is now acceptable for publication, you may indicate that here to bypass the “Comments to the Author” section, enter your conflict of interest statement in the “Confidential to Editor” section, and submit your "Accept" recommendation.

Reviewer #2: All comments have been addressed

Reviewer #3: (No Response)

2. Is the manuscript technically sound, and do the data support the conclusions?

Reviewer #2: Yes

Reviewer #3: Partly

3. Has the statistical analysis been performed appropriately and rigorously? 

Reviewer #2: I Don't Know

Reviewer #3: Yes

4. Have the authors made all data underlying the findings in their manuscript fully available?

Reviewer #2: Yes

Reviewer #3: Yes

5. Is the manuscript presented in an intelligible fashion and written in standard English?

Reviewer #2: Yes

Reviewer #3: (No Response)

6. Review Comments to the Author

Reviewer #2: (No Response)

Reviewer #3: (No Response)

7. PLOS authors have the option to publish the peer review history of their article (what does this mean?). If published, this will include your full peer review and any attached files.

Reviewer #2: No

Reviewer #3: No

---

## [Author Response · Author response to Decision Letter 1]

2 Apr 2024

Dear Editor:

We carefully studied the valuable comments of the reviewers and tried our best to revise the manuscript titled “Identification of ADAMTSL2 as a potential prognosis biomarker associated with immune infiltrates and drug sensitivity in colorectal cancer” (PONE-D-23-31014R1). The amendment has been marked with red. We have made detailed changes throughout the text. The point-to-point responses to the reviewer’s comments are listed in the revised text.

Comments 

This is an interesting manuscript that explores the impact of ADAMTSL2 as a potential prognostic biomarker in colorectal cancer. By using database, the authors conclude that the high expression levels of ADAMTSL2 is associated with poor prognosis in CRC. Although the dataset is huge und the fidings are impressive, the study lacks experimental validation that do not allow to draw strong conclusions at this point.

1. The analysis of the database in the mauscript are strong and the most conclusions are based on these analysis. Key findings should, however, be validated by other methodology. For instance, while the authors claim that ADAMTSL2 expression is highe in CRC tissues compared to normal and paired normal area, assessments on human samples should be performed.

Response: Thanks for your comments. We investigated the prognostic value of ADAMTSL2 in CRC using TCGA-CRC data and validated the expression of ADAMTSL2 using other clinical cohort datasets GSE71187 (157 CRC tissues and 32 normal tissues). At present, our experimental funding is limited. We will actively apply for research funding based on existing research results. Once funding support is obtained, we will conduct more detailed research in the future. We mention this limitation in the article. This study is a clinical significance and regulatory network analysis of ADAMTSL2 in CRC using public databases of TCGA and GEO, and these results must be further validated in real-world samples.

2. All abbreviations should be fully described when they appear for the first time, such as TGF-beta, ECM-receptor and so on. And once the word is described with full description, it should not be repeated. The authors should completely go through the paper again.

Response: Thanks for your comments. We carefully reviewed the whole text according to your suggestions and made necessary modifications.

3. There are several sentences containing the phrase “and others”. This is not scientific writing and should be revised.

Response: Thanks for your comments. The sentences were improved according to your suggestion.

4. The authors claim that ADAMTSL2 is associated with immune infiltration but this cannot be concluded only with the database analysis. Either the statement in accordance with the title of the manuscript should be reviced or the validation assessments using human samples should be conducted.

Response: Thanks for your comments. We have adjusted the title of the manuscript according to your suggestion to " ADAMTSL2 is a novel prognostic biomarker in colorectal cancer".

Journal Requirements:

Response: Thanks for your comments. We have carefully checked all the references according to your suggestion. We confirm that the reference list is complete and accurate.

Thank you for considering our manuscript.

Yours sincerely,

Zhe Huang

---

## [Editor Report · Decision Letter 2]

9 Apr 2024

PONE-D-23-31014R2ADAMTSL2 is a novel prognostic biomarker in colorectal cancerPLOS ONE

Dear Dr. Huang,

Thank you for submitting your manuscript to PLOS ONE. After careful consideration, we feel that it has merit but does not fully meet PLOS ONE’s publication criteria as it currently stands. Therefore, we invite you to submit a revised version of the manuscript that addresses the points raised during the review process.

The lack of validation in real samples is the weak point of your article. However, I think ADAMTSL2 should be investigated as a potential marker in immuno-oncology. Please reconsider the title (delete novel) and include some information that these are from dataset analysis.

We look forward to receiving your revised manuscript.

Kind regards,

Shiki Fujino, M.D.

Academic Editor

PLOS ONE
---

## [Author Response · Author response to Decision Letter 2]

10 Apr 2024

Dear Editor:

We carefully studied the valuable comments of the editor and tried our best to revise the manuscript titled “ADAMTSL2 is a novel prognostic biomarker in colorectal cancer” (PONE-D-23-31014R2). The amendment has been marked with red. 

Comments 

The lack of validation in real samples is the weak point of your article. However, I think ADAMTSL2 should be investigated as a potential marker in immuno-oncology. Please reconsider the title (delete novel) and include some information that these are from dataset analysis.

Response: Thank you for your comments. We investigated the prognostic value of ADAMTSL2 in CRC using TCGA-CRC data and validated the expression of ADAMTSL2 using other clinical cohort datasets GSE71187 (157 CRC tissues and 32 normal tissues). At present, we do not have funding for experiments in the real world. We will actively apply for research funding based on existing research results. Once funding support is obtained, we will conduct more detailed research in the future. We mention this limitation in the article. This study is a clinical significance and regulatory network analysis of ADAMTSL2 in CRC using public databases of TCGA and GEO, and these results must be further validated in real-world samples.

The title has been adjusted to " ADAMTSL2 is a potential prognostic biomarker and immunotherapeutic target for colorectal cancer: bioinformatic analysis and experimental verification" according to your suggestion.

Thank you for considering our manuscript.

Yours sincerely,

Zhe Huang

---

## [Editor Report · Decision Letter 3]

3 May 2024

ADAMTSL2 is a potential prognostic biomarker and immunotherapeutic target for colorectal cancer: bioinformatic analysis and experimental verification

PONE-D-23-31014R3

Dear Dr. Huang,

We’re pleased to inform you that your manuscript has been judged scientifically suitable for publication and will be formally accepted for publication once it meets all outstanding technical requirements.

Kind regards,

Shiki Fujino, M.D.

Academic Editor

PLOS ONE
---

## [Editor Report · Acceptance letter]

9 May 2024

PONE-D-23-31014R3 

PLOS ONE

Dear Dr. Huang, 

I'm pleased to inform you that your manuscript has been deemed suitable for publication in PLOS ONE. Congratulations! Your manuscript is now being handed over to our production team.

Kind regards, 

on behalf of

Dr. Shiki Fujino 

Academic Editor

PLOS ONE